# Impact of the Maximum Number of Switching Reconfigurations on the Cost Saving in Network Function Virtualization Environments with Elastic Optical Interconnection †

**Vincenzo Eramo [1]** , **Francesco G. Lavacca [2,*]** and **Tiziana Catena [1]**

[1]   DIET, "Sapienza" University of Rome, Via Eudossiana 18, 00184 Rome, Italy;
    vincenzo.eramo@uniroma1.it (V.E.); tiziana.catena@uniroma1.it (T.C.)
[2]   Fondazione Ugo Bordoni, Viale del Policlinico 147, 00161 Roma, Italy
*   Correspondence: fglavacca@fub.it
†   This paper is an extended version of paper titled *Proposal and Investigation of an Optical Reconfiguration Cost Aware Policy for Resource Allocation in Network Function Virtualization Infrastructures* and presented in the 21st International Conference on Transparent Optical Networks (ICTON) held in Angers, France, 9–13 July 2019.

**Abstract:** Network Function Virtualization is based on the virtualization of the network functions and it is a new technology allowing for a more flexible allocation of cloud and bandwidth resources. In order to employ the flexibility of the technology and to adapt its use according to the traffic variation, reconfigurations of the cloud and bandwidth resources are needed by means of both migration of the Virtual Machines executing the network functions and reconfiguration of circuits interconnecting the Virtual Machines. The objective of the paper is to study the impact of the maximum number of switch reconfigurations on the cost saving that the Networking Function Virtualization technology allows us to achieve. The problem is studied in the case of a scenario with an elastic optical network interconnecting datacenters in which the Virtual Machines are executed. The problem can be formulated as an Integer Linear Programming one introducing a constraint on the maximum number of switch reconfigurations but due to its computational complexity we propose a low computational complexity heuristic allowing for results close to the optimization ones. The results show how the limitation on the number of possible reconfigurations has to be taken into account to evaluate the effectiveness in terms of cost saving that the Virtual Machine migrations in Network Function Virtualization environment allows us to achieve.

**Keywords:** Network Function Virtualization; elastic optical network; cloud and bandwidth resource; switch reconfiguration

## 1. Introduction

The Network Function Virtualization (NVF) technology, [1] based on the principles of virtualization and service-orientation, has deeply changed the operation mode of the telecommunication networks [2]. The NFV paradigm introduces the Service Function Chain (SFC) that is a set of Service Functions to be executed according to a given order for a traffic flow. The service delivery involves the allocation to the SFC of both cloud and bandwidth resources. The first ones are needed to provide memory, processing, disk resources to the Virtual Machines (VM) executing the Service Functions (SFs) and located in datacenters; the second ones are needed to interconnect VMs executing the SFs of a SFC.

In recent years, some researches have been done on network planning problems in NFV environment, which mainly focused on the SFC routing and Virtual Network Function (VNF)

deployment problem (e.g., [3–6]). To minimize the cost of the VNF deployment in an NFV datacenter environment with Optical Network interconnection [7–9] , an integer linear programming model was first established, and then three heuristic algorithms were proposed to solve the integer linear programming model effectively [10]. To minimize the total cost of the energy consumption and the revenue loss due to QoS degradation, an efficient algorithm, which is based on the back-to-back strategy, was designed [11]. To solve service chain and resource allocation problem, a mixed-integer linear programming model was proposed [5], and a heuristic-based algorithm, which consists of two sub-algorithms: one-hop optimal traffic scheduling algorithm and VNF chain composition algorithm, was proposed. Li et al. [12] formulated the VNF deployment problem as an integer linear programming model and proposed a simulated annealing based heuristic to get approximate solutions in shorter time. Marouen et al. [13] proposed an algorithm based on eigenvalue decomposition for the VNF deployment in EON (Elastic Optical Network) for the sake of satisfying user requirements and maximizing provider revenue. To lower the costs, reconfiguration of cloud and bandwidth resources is needed. Reconfigurations algorithms [14] have been proposed that move the execution of the VMs so as to guarantee the use of lower cost datacenters. Though the reconfiguration policies allows for lower costs, they involve reconfiguration costs that have been investigated in [14]. In the case of optical interconnection of the NFV sites, the high optical switch reconfiguration time leads to limit the number of switch reconfigurations to guarantee high switch throughput [15]; for instance, the circuit reconfiguration time of a Micro-ElectroMechanical Systems (MEMS) based optical crossconnect is in the order of 1 s [15] that leads to no more than one reconfiguration every 10 s to reach 90% switch throughput. The objective of the paper is to evaluate the advantages that the NFV technology with optical interconnection allows us to achieve in terms of cloud and bandwidth resource cost saving under the constraint on the maximum number of switch reconfigurations. We have provided an Integer Linear Programming (ILP) formulation of the optimization problem. It was inherited from the one proposed in [16] with the addition of a constraint on the maximum number of switch reconfigurations. Because of the computational complexity of the defined ILP, we propose a low computational complexity heuristic that allow us to study very complex network scenarios. The heuristic is based on the application of a constrained shortest path algorithm proposed in literature and referred to as Label Setting [17,18]; it consists of the evaluation of a shortest path in a graph in which each edge is labeled with two labels; the shortest paths are evaluated on the basis of the values of the first labels while the second label allows us to take into account the constraint on the maximum number of switch reconfigurations.

The paper is organized in five Sections. The related work is reported in Section 2. The reference scenario is described in Section 3. An efficient heuristic is illustrated in Section 4. The results of the heuristic and the optimization ones are compared in Section 5 in the case of a small size network. The impact of the maximum number of switch reconfigurations on the cost saving is also evaluated in Section 5 in the case of large networks. Conclusions and future research items are discussed in Section 6. Appendix A provides an Integer Linear Programming (ILP) formulation of the cloud and bandwidth resource allocation problem when the number of switch reconfigurations is limited. Appendix B provides a formal description of the proposed heuristic.

## 2. Related Work

The traffic variations and the flexibility of the NFV technology involve resource configurations during the lifetime of a SFC. For example when the traffic decreases, algorithms can be applied in order to consolidate the used resources [19]. The algorithms involve the VM migration towards fewer datacenter and the switching off of the remaining datacenters with advantages in terms of cloud resource costs. To support the VM migration, network paths also need to be reconfigured in order to interconnect the migrated VMs. The reconfiguration cost has been investigated in some research works. It may be characterized by: 1) the gain loss of a network operator due to the information loss during the VM migration and the circuit reconfiguration [20]; the energy consumption occurs when the VM memory content is moved from a server to another [11]. The problem has also been investigated

in [16,21] in the case in which the datacenters are interconnected by an EON; the reconfiguration cost is characterized by the lost data during the reconfigurations of optical devices in the periods in which VM migrations are needed. Differently from electronic switches, the optical ones, though they have the potential to reduce the overall cost and the power consumption, allow for a lower reconfiguration rate dependently on the used technology. Two main technologies are actually used: i) the 3D-MEMS-based cross-connects [15,22,23] that perform transparent optical switching using microscopic tiltable mirrors to reflect light from the input fibers to the input fibers; ii) the Reconfigurable Optical Add Drop Multiplexers (ROADMS) [15,24–26] that are often based on 2D-MEMS allowing then to switch more quickly than 3D MEMS. The ROADMs have the advantage of low switching times in the order of 10 μ*s* but they allow only for the realization of small size optical switches and consequently they are not suitable to interconnect NFV sites. Conversely, 3D-MEMS based cross-connects are cost effective, allow for the realization of high size switches but they have the drawback of slow reconfiguration times, typically ranging from 10 ms to 1 s. [15]. Furthermore when a reconfiguration is performed, the traffic of the reconfigured circuit is lost. The consequence is that, in order to maintain high the average switch throughput, the reconfiguration rate has to be maintained low. For instance, if the reconfiguration time is $\sigma$, a given circuit configuration has to be held at least for a time of $9\sigma$ to get a 90% throughput. That leads to maximum reconfiguration rates of the 3D-MEMS based cross-connects ranging from 100 ms to 10 s. The objective of the paper is to study the impact of the limited reconfiguration rate of the optical switches on the cost saving that reconfiguration policies allow us to obtain in NFV architectures where the NFV sites are optically interconnected. We formulate the optimization problem and due to its complexity we propose a heuristic based on the evaluation of a constrained shortest path [27] in a multi-stage graph.

## 3. Reference Scenario

NFV is a new paradigm that employs cloud infrastructures to support telecommunication services. The service functions, also referred to as VNF are executed in VM hosted in geographically-distributed data-centers referred to as NFV Infrastructure-Point-of Presence (NFVI-PoP) and managed by different providers. To support high bandwidth service requests, we consider an Elastic Optical Networks (EON) to provide optical circuits interconnecting the VM located in different NFVI-PoPs.

Due to the dynamic nature of the applications, the traffic flowing through VM can change over time. To reduce the costs in dynamic traffic scenario, we propose a solution to maintain as low as possible the processing and bandwidth costs, changing over time the NFVI-PoPs in which the VMs are executed. Given that the migration of the VMs across the network may lead to many optical circuit reconfigurations and consequently to bit loss and revenue loss of the Network Operator (NO), we also take into account the number of reconfigurations to be applied to the elastic optical network. A possible reconfiguration example from an high to a low traffic scenario is reported in Figure 1a, where two different NFVI-PoPs host a Firewall (FW) and an Intrusion Detection System (IDS) VMs exploited to satisfy the SFC request reported in Figure 1. The black circles represent the cores required from the VM.

In particular, notice how in the low traffic interval, the FW VM can be moved from NFVI-PoP #1 to NFVI-PoP #2 because the cloud resources of NFVI-PoP #2 are less expensive than the ones of NFVI-PoP #1. This migration leads to tear down the two lightpaths between the access node E1 and the NFVI-PoP #1 and between NFVI-PoP #1 and NFVI-PoP #2, reported with dashed lines in Figure 1a. As shown in Figure 1b a new lightpath is then set up between access node E1 and NFVI-PoP #2, while the ligthpath represented by the continuous line remains the same.

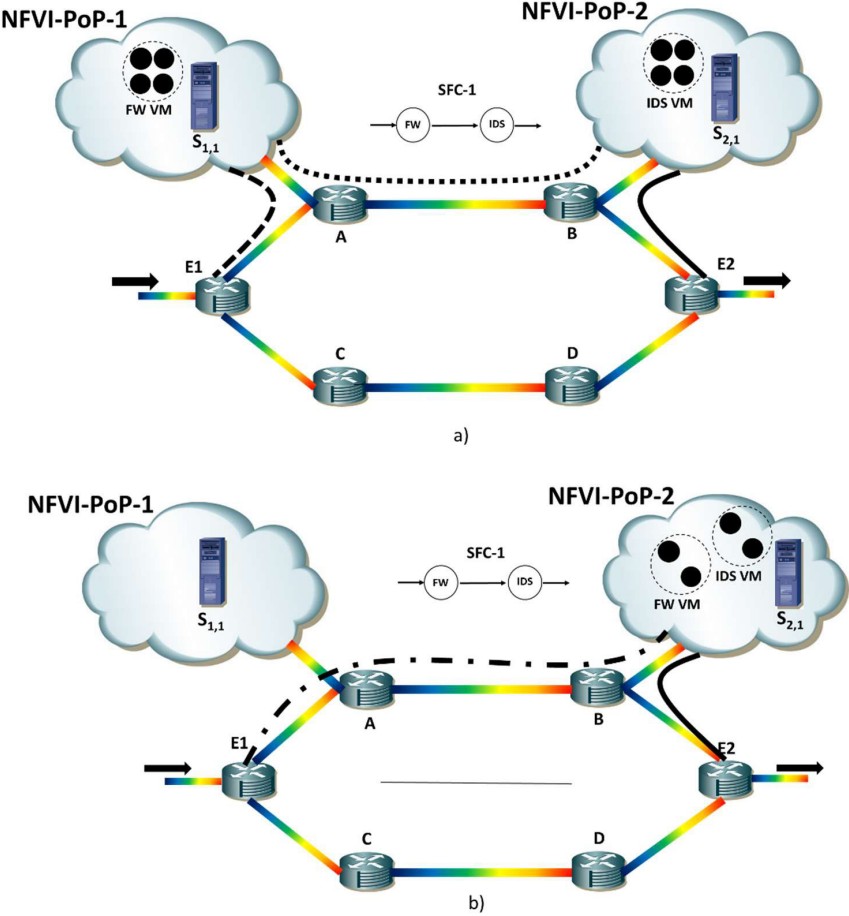

**Figure 1.** Interconnection of NFV Infrastructure-Points-of Presence (NFVI-PoPs) by means of an Elastic Optical Network (**a**); Migration of a Virtual Machine (VM) from NFVI-PoP-1 to NFVI-PoP-2 (**b**).

The proposed solution is based on:

- the application of a VM consolidation technique towards the less costly NFVI-PoPs;
- the agility of an elastic optical network in reconfiguring optical lightpaths between active NFVI-PoPs.

The dynamic allocation problem of processing and bandwidth resource consists in determining:

- in each stationary traffic interval both the NFVI-PoPs to be held active and the processing core amount to be allocated to each VM;
- the lightpaths to be reconfigured, the more appropriate modulation index according to the optical path length and the maximum number of switch reconfigurations.

The objective is to minimize the total cost given by the sum of the processing and bandwidth cost with the constraint of not exceeding the maximum number of switch reconfigurations. In some cases, this constraint may limit the number of VMs to be migrated in order to reduce the cost to be paid by the Network Operator because of QoS degradation occurring during the VM migrations.

We assume a cycle-stationary traffic con $T$ Stationary Intervals (SI). We have formulated the ILP problem. It was inherited from the one proposed in [16] with the addition of a constraint on the maximum number $R$ of switch reconfigurations. The output of the ILP provides for each SI: (i) in which NFVI-PoPs the VMs are allocated; (ii) the cloud resources, in terms of number of cores, allocated to each VM; (iii) the optical paths connecting each tuple of VMs, as well as the modulation system (BPSK, QPSK, 16QAM) used. The objective function to be minimized is characterized by the sum of cloud and bandwidth resource costs. Due to the high complexity of the optimization problem, we have defined a heuristic that allows us to achieve a low computational complexity reconfiguration policy.

## 4. Label Setting Based Heuristic

We propose a two-steps solution to solve the cloud resource and bandwidth reconfiguration problem with computational complexity as low as possible under the constraint of the maximum number of optical switch reconfigurations.

The section is organized in two Subsections. The main notations are reported in Section 4.1 while the operation mode of the proposed heuristic is reported in Section 4.2.

### 4.1. Notations

We report in Table 1 the main notations. Let us represent the cloud/network infrastructure with the graph $G_o = (V_s, L_o)$ where $V_s$ is the node set and $L_o$ is the network link set. $V_s$ is given by the union of the sets $V_o$ and $V_{NFVI-PoP}$. $V_o$ is the set of optical switches while $V_{NFVI-PoP}$ is the set of NFVI-PoPs in which the VMs are executed. The main parameters of the optical network and their definitions are the following: $L_e$ is the fiber length of the link $e \in L$; $c_b$ is the bandwidth cost unit expressed in ($\$/GHzKmhour$); $\Psi$ is the set of the network paths; $B_{FS}$ is the bandwidth (Hz) of a Frequency Slot (FS).

**Table 1.** NFVI-PoP Infrastructure and network Parameters.

| Parameter | Definition |
|---|---|
| $V_o$ | Set of optical switches |
| $V_{NFVI-PoP}$ | Set of NFVI-PoPs |
| $L_o$ | Set of Network Links |
| $L_e$ | Length of the link $e \in Ł_o$ |
| $c_b$ | Bandwidth Cost Unit |
| $\Psi$ | Set of network paths |
| $B_{FS}$ | Bandwidth of a Slot Frequency |
| $F$ | Number of Service Functions (SF) types that can be executed in the NFVI-PoP |
| $C_{p,h}$ | Maximum processing capacity of the i-*th* type VM |
| $n_h^c$ | Number of cores to be allocated to a i-*th* type VM |
| $c_v^{core}$ | Cost unit of renting for one hour one processing core in the NFVI-PoP $v \in V_{NFVI-PoP}$ |
| $V_{VM}$ | Set of VMs |
| $L_v$ | Set of Virtual Links |
| $f_v(S)$ | Bandwidth requested by the virtual node $v \in V_{VM}$ in the traffic state $S$ |
| $f_l(S)$ | Bandwidth requested by the virtual link $l \in L_v$ in the traffic state $S$ |

The VMs supporting the Service Functions (SF) are executed in the NFVI-PoPs. We assume that $K$ types of VMs can be executed according to the corresponding SF. The type $h$-th ($h = 1, \cdots, K$) VM is characterized by the maximum processing capacity $C_{p,h}$ ($h = 1, \cdots, K$) and the allocation of $n_h^c$ ($h = 1, \cdots, K$) cores. We also assume that the VMs can be migrated between NFVI-PoPs and that, when the traffic decreases, vertical scaling techniques can be applied by linearly reducing the VM processing capacity and proportionally the number of allocated cores. We also introduce the parameter $c_v^{core}$ defined as the cost of renting a VM for one hour in the NFVI-PoP $v \in V_{NFVI-PoP}$.

The support of a given number of Service Function Chains (SFC) involve the construction of the Virtual Network Function Forwarding Graph (VNFFG) $G_v = (V_{VM}, L_v)$ [28], a virtual graph where $V_{VM}$ is a set of VMs to be instantiated in NFVI-PoPs and $L_v$ is a set of virtual links to be supported by network paths. Let $\tau_{h,v}$ be a binary parameter assuming the value 1 if the VM $v \in V_{VM}$ is of $k$-th type, otherwise its value is 0. In the traffic state $S$ the VNFFG is characterized by the following parameters: $f_v(S)$ is the bandwidth requested by the virtual node $v \in V_{VM}$ in the traffic state $S$; $f_l(S)$ is the bandwidth requested by the virtual link $l \in L_v$ in the traffic state $S$.

The proposed heuristic has the objective of evaluating in each traffic state $S$ an embedding $\Gamma$ that determines in which NFVI-PoPs and network paths, the nodes $v \in V_{VM}$ and link $l \in L_v$ have to be supported respectively. The embedding is characterized by the following parameters: $x_{vw}^\Gamma(S)$ assumes the value 1 if the VM $w \in V_{VM}$ is executed in the NFVI-PoP $v \in V_{NFVI-PoP}$, otherwise its value is 0;

$y_{pl}^{\Gamma}(S)$ assumes the value 1 if the virtual link $l \in L_v$ is routed on the network path $p \in \Psi$, otherwise its value is 0.

The choice of one embedding leads to the consumption of cloud and bandwidth resources. The embedding is characterized by a cost $C^{\Gamma}(S)$ given by the sum of the cloud cost $C_{cl}^{\Gamma}(S)$ and bandwidth cost $C_{bw}^{\Gamma}(S)$. According to the above introduced parameters, we can express $C_{cl}^{\Gamma}(S)$ and $C_{bw}^{\Gamma}(S)$ as follows:

$$C_{cl}^{\Gamma}(S) = T_s \sum_{v \in V_{NFVI-PoP}} c_v^{core} \sum_{w \in V_{VM}} x_{vw}^{\Gamma}(S) \left\lceil \frac{f_v(S)}{\sum_{h=1}^{K} \tau_{h,v} \frac{C_{p,h}}{n_h^c}} \right\rceil \tag{1}$$

$$C_{bw}^{\Gamma}(S) = T_s c_b \sum_{l \in L_v} \sum_{p \in \Psi} \sum_{e \in L_o} B_{FS} S_{l,p}(S) L_e \delta_{e,p} \tag{2}$$

where $T_s$ is the duration of the traffic state $S$, $\delta_{e,p}$ is a binary parameter assuming the value 1 if the link $e \in L_o$ belongs to the path $p$, otherwise it is 0. $S_{l,p}(S)$ is the number of FSs to be allocated on the path $p$ to support the virtual link $l$; $S_{l,p}$ depends on the bandwidth $f_l(S)$ requested by the link $l$ in the traffic state $S$ and the path length that determines the most suitable modulation system (BPSK, QPSK, 8QAM, 16QAM).

The change of a traffic state may lead to an embedding change that needs lightpath changes and the reconfiguration of optical switches. The proposed heuristic aims at choosing embeddings that minimize the cloud and bandwidth costs and under the constraint to limit the number of optical switch reconfigurations. In particular when the traffic state changes from $S_i$ to $S_j$ and the embedding changes from $\Gamma_i$ to $\Gamma_j$, the number $R(\Gamma_i, S_i, \Gamma_j, S_j)$ of optical switch reconfigurations can be expressed as follows:

$$R(\Gamma_i, S_i, \Gamma_j, S_j) = \sum_{l \in V_{VM}} \sum_{e \in L_o} (1 - \rho_{l,e}) \tag{3}$$

where $\rho_{l,e}$ is a variable assuming the value 1 if the virtual link $l$ has changed the occupied resources in the optical link $e$ after the embedding change.

We provide an Integer Linear Programming (ILP) formulation of the cloud and bandwidth resource allocation problem in Appendix A. The problem is inherited from the one proposed in [16] with a more simple formulation and the addition of the constrain on the maximum number of switch reconfigurations.

## 4.2. Heuristic Description

Given the complexity of the optimization problem, which allows us to solve the problem only for small network scenarios, we defined a heuristic composed of two procedures applied in sequence. A first procedure, called NFV/Optical Resource Reconfiguration (NORR) heuristic, and a second procedure, called Optical Network Remapping Costs and Reconfigurations Aware (ONRCRA) heuristic.

The objective of the NORR heuristic, described in [16], is to find in each h-th ($h = 0, \cdots, T-1$) SI an embedding $\Gamma^{(h)}$ of the VNFFG, so that the total processing and bandwidth cost $C^{(h)}$ is minimized. In this first phase, the reconfiguration costs of transition from one embedding to another are not taken into account.

The ONRCRA heuristic takes the embeddings of the set $\Theta = \{\Gamma^{(0)}, \Gamma^{(1)}, \cdots, \Gamma^{(T-1)}\}$ calculated by the NORR procedure, and creates a multistage graph, of which two examples are given in Figure 2a,c. Each node of the graph represents an embedding belonging to the set $\Theta$ that is admissible for that particular SI, and is labeled with the associated cost, calculated in the first step. Each edge represents the transition from one embedding to another and is labeled with the number of reconfigurations to be performed on the optical switches, whose value is calculated by ONRCRA heuristic.

Once the multistage graph has been created, the final objective of the ONRCRA heuristic is to calculate on it the sequence $\Sigma^{opt} = \{\Psi_0, \Psi_1, \cdots, \Psi_{T-1}\}$ of embeddings to be applied in the entire time interval, such that the total cost (which is the sum of the costs of the chosen embeddings) is minimized, but not exceeding the preset number of reconfigurations. In finding the solution, therefore, the ONRCRA heuristic will take into account that the transition from a certain embedding to another could cause many reconfigurations of the optical paths and therefore a high bit loss; for this reason, embeddings with the lowest cost will not necessarily be selected in the final solution.

The resolution of this problem is identified in literature as Weight Constrained Shortest Path Problem (WCSPP) [17]: given a directed graph which has cost and a weight associated to each edge, the WCSPP consists of finding the least-cost path between two specified nodes, such that the total sum of weights is less than a specified value.

The algorithm chosen for the WCSPP solution is the Label Setting Algorithm [17], an implicit enumeration algorithm which leads to an optimal solution, generalized from the one given by Dijkstra. Its name derives from the fact that it is an algorithm that uses labels to iteratively explore the graph keeping track of the possible paths that respect the constraint on the maximum number of reconfigurations. Each label contains two values: the total cost, equal to the sum of the costs of the embeddings selected up to that point, and the number of reconfigurations, the sum of all the reconfigurations determined by the path considered. Starting from a single label in the first node, at each iteration the minimum-cost label is expanded in the direction of the next SI, and the procedure is repeated until the last SI is reached; if the number of reconfigurations exceeds the maximum allowed value, the label is discarded (and along with it the associated path). The label that reaches one of the nodes of the last SI, will be the one that identifies the path at minimum cost and such that the constraint on the number of reconfigurations is respected.

Two examples of execution of the Label Setting algorithm are described for the multi-stage graphs illustrated in Figure 2a,c respectively. The steps of the algorithm are described in Figure 2b,d respectively. The darker label is the one selected to be extended in next iteration of the algorithm and the thick lines are the directions of extension in the graph. The maximum number of switch reconfigurations is assumed to be 10 for both scenarios. Note that in while in step 2 of Figure 2b the label chosen to be expanded is the one with the least cost, in step 2 of Figure 2d the label (65,15) has to be discarded because the constraint on the maximum number of reconfigurations is violated.

The solution paths are highlighted by the edges in bold in Figure 2a,c respectively. A pseudo-code of the proposed heuristic is reported in Appendix B.

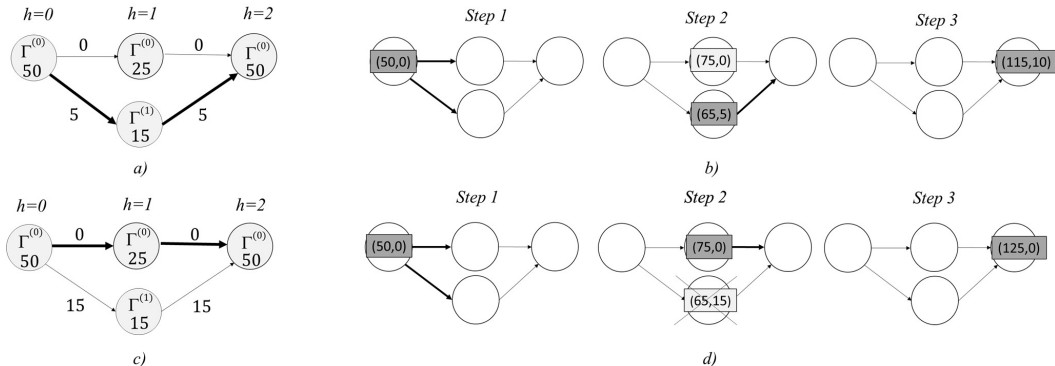

**Figure 2.** Two examples of application of the Label Setting algorithm. The initial multi-stage graphs are reported in (**a**,**c**) for different reconfiguration costs; the procedures for the evluation of the constrained shortest path are reported in (**b**,**d**) for the multi-stage graphs (**a**,**c**) respectively.

## 5. Numerical Results

We evaluate the effectiveness of the proposed heuristic for the networks of Figures 3–5. The network of Figure 3 is composed by four access nodes, four optical switches and four NFVI-PoPs;

its small size allows us to compare the results of the heuristic to the ones of the ILP problem inherited from [16] with the addition of a constraint on the maximum number of switch reconfigurations. The Deutsche Telekom network of Figure 4 is composed by fourteen access nodes, fourteen optical switches and four NFVI-PoPs located in the nodes of Leipzig, Hannover, Frankfurt and Nuremberg. The USNET network is composed by twenty-four access nodes, twenty-four switches and four NFVI-PoPs connected to the switches S1, S2, S3 and S4. The Deutsche Telekom and USNET network scenarios will allow us to evaluate the impact of the limited number of switch reconfigurations on the cost saving in realistic case studies. The fiber lengths are reported as labels in Figures 3–5.

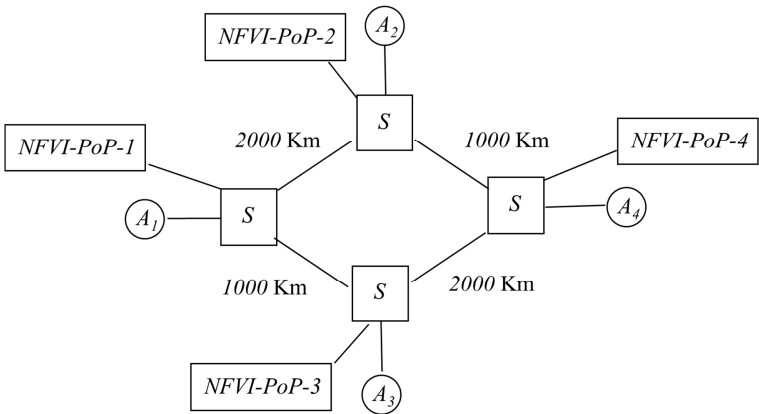

**Figure 3.** Optical Network Topology with four access nodes, four optical switches and four NFVI-PoPs.

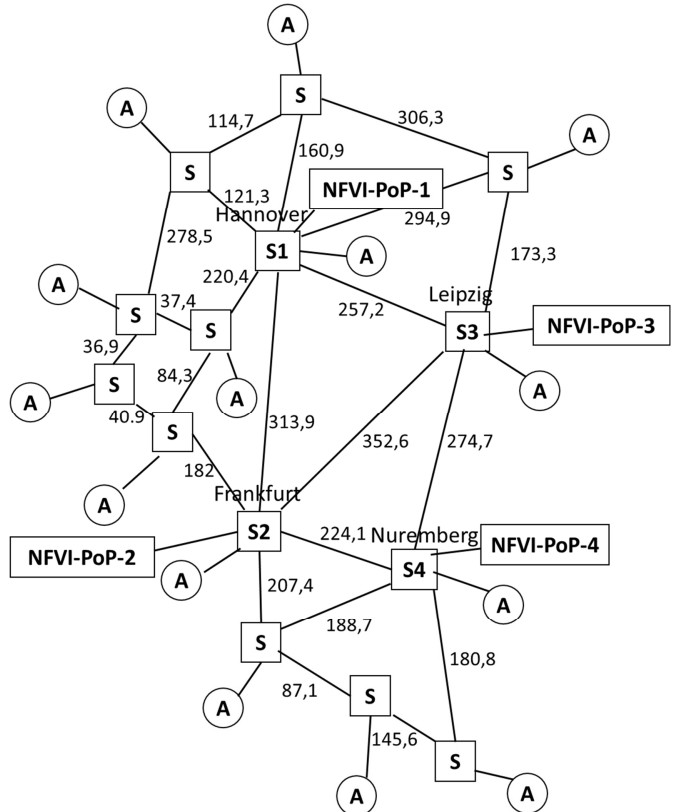

**Figure 4.** Deutsche Telekom Network.

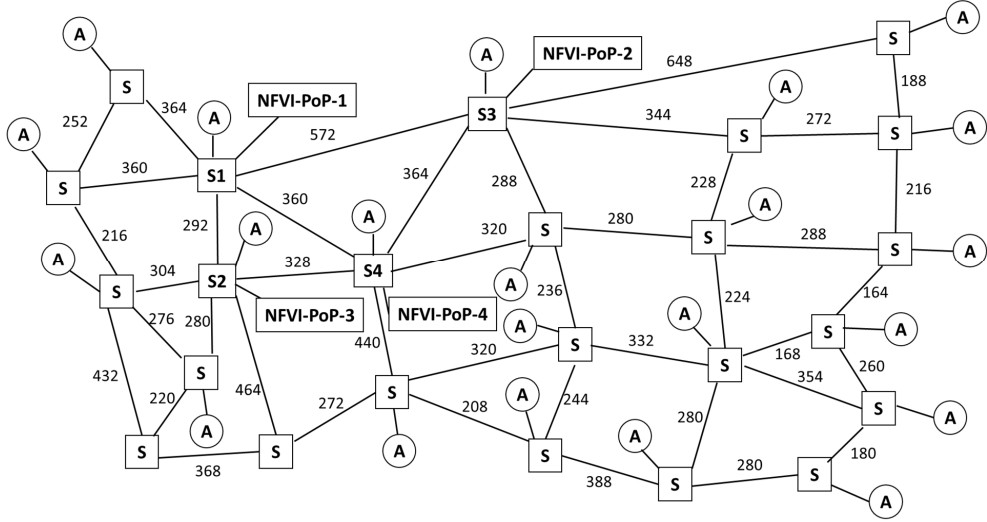

**Figure 5.** USNET Network.

The spectrum in each fiber is organized in frequency slot of 6.25 GHz. We assume that adaptive modulation systems are used whose choice determines the maximum path length. In particular the BPSK, QPSK, 8QAM and 16QAM modulations are used with path length $L_{BPSK}$, $L_{QPSK}$, $L_{8QAM}$ and $L_{16QAM}$ reported in Table 2.

**Table 2.** Maximum Path Length for the Binary Phase Shift Keying (BPSK), Quadrature Phase Shift Keying (QPSK), 8 Quadrature Amplitude Modulation (8QAM) and 16QAM modulation systems.

| | |
|---|---|
| $L_{BPSK}$ | 3000 Km |
| $L_{QPSK}$ | 1500 Km |
| $L_{8QAM}$ | 750 Km |
| $L_{16QAM}$ | 375 Km |

The NFVI-PoPs are equipped with four types of VMs executing the Firewall (FW), Intrusion Detection System (IDS), Network Address Translator (NAT) and Proxy service functions. The number $n^c$ of cores to be allocated to the various types of VMs and their maximum processing capacity $C_p$ are reported in Tables 3 and 4 for the NFVI-PoPs located in the networks of Figures 3 and 4 respectively.

**Table 3.** Maximum processing capacity $C_p$ and allocated number $n^c$ of cores for the VMs implementing Firewall (FW), Intrusion Detection System (IDS), Network Address Translator (NAT) and Proxy in the network scenario of Figure 3.

| Parameter | $C_p$ | $n^c$ |
|---|---|---|
| **FW** | 60 Gbps | 260 |
| **IDS** | 60 Gbps | 790 |
| **NAT** | 60 Gbps | 130 |
| **PROXY** | 60 Gbps | 260 |

**Table 4.** Maximum processing capacity $C_p$ and allocated number $n^c$ of cores for the VMs implementing FW, IDS, NAT and Proxy in the network scenario of Figure 4.

| Parameter | $C_p$ | $n^c$ |
|---|---|---|
| **FW** | 30 Gbps | 130 |
| **IDS** | 30 Gbps | 350 |
| **NAT** | 30 Gbps | 65 |
| **PROXY** | 30 Gbps | 130 |



To obtain the results of the heuristic we have implemented it in a JAVA code. Conversely the results of the optimization problem have been achieved by using the CPLEX tool. First we carry out a comparison between the results of the heuristic and the ILP problem for the small network scenario of Figure 3. We assume that each NFVI-PoP is equipped with a total number of 2000 cores. The core cost is equal to 0.74 $/h, 0.89 $/h, 1.08 $/h and 1.29 $/h for the NFVI-PoP-1, NFVI-PoP-2, NFVI-PoP-3 and NFVI-PoP-4 respectively [28]. The total number of frequency slots in each optical fiber is 50. The bandwidth cost is equal to 0.001 cent/KmGHzhour. We assume a cycle-stationary traffic [16] with two only SIs. The total offered in the peak SI is 200 Gbps and a 80% traffic reduction occurs in the other SI. Four type of SFCs are considered: the first one composed by a FW, the second one is composed by a FW and an IDS, the third one composed by a FW, an IDS and a NAT and the fourth one is composed by a FW, an IDS, a NAT and a PROXY. The SFC required bandwidth in the peak SI is chosen according to a ZIP distribution among the values 1 Gbps, 1.5 Gbps, 2 Gbps, 2.5 Gbps and 3 Gbps. We report in Table 5 the total cost as a function of the maximum number of reconfigurable lightpaths when the proposed heuristic is applied and the ILP problem described in Appendix A is solved. The percentage error is also evaluated. From the values reported in Table 5 we can observe how in the case of small networks the heuristic allows for percentage errors lower than 25%. Unfortunately the high running times of the optimization problem does not allow for analysis of more complex scenario.

**Table 5.** Comparison between cost of the Integer Linear Programming (ILP) and heuristic as a function of the maximum number $R$ of switch reconfigurations.

| $R$ | 0 | 100 | 200 | 300 | 400 | 500 |
|---|---|---|---|---|---|---|
| ILP | 172243 | 172243 | 172243 | 170710 | 159959 | 158465 |
| Heuristic | 131925 | 131190 | 130958 | 130958 | 130958 | 130958 |
| Percentage Error | 23% | 24% | 24% | 23% | 18% | 17% |

For this reason we only provide the results of the heuristic in the network scenario of Figure 4. The total number of cores in each NFVI-PoP is 4092. We assume a core cost unbalancing for the NFVI-PoP-1, NFVI-PoP-2, NFVI-PoP-3 and NFVI-PoP-4. The expression of the core cost for the four NFVI-PoPs is $4c_{av}\frac{1-\alpha}{1-\alpha^4}\alpha^i$ ($i = 0, 1, 2, 3$) where $\alpha$ is the cost unbalancing factor and $c_{av}$ is the average core cost. $c_{av}$ is assumed to be equal to 1$/h. The choice of $\alpha = 1$ corresponds to the uniform cost case.

The bandwidth cost is 0.001 cent/KmGHzhour and the number of frequency slots in each fiber is 600. The SFC type and their bandwidth are chosen as in the small network scenario. A classical daily profile is considered with 8 SIs. The total offered traffic is 800 Gbps in the peak SI. The traffic reduction in the low traffic SI is 80% the one in the peak SI. We report the total cost, the processing cost, the bandwidth cost and the number of reconfigured switches as a function of the maximum number $R$ of switch reconfigurations in Figure 6a–d respectively. We evaluate the performance index as a function of the parameters $R$ ranging from 0 to 20,000 in order to investigate the performance in the case of optical switches with different reconfiguration characteristics.

The cost unbalancing parameter $\alpha$ is varied from 1 to 1.5. From the Figure 6 we can remark that:

- the increase in maximum number $R$ of switch reconfigurations leads to lower total costs up a saturation value depending on the chosen value of $\alpha$ (Figure 6a);
- the cost advantage is due to the consolidation of the VMs toward less costly NFVI-PoPs and consequently to lower processing costs (Figure 6b);
- the bandwidth cost has little impact on the total cost for the typical choice of the bandwidth cost per GHz and Km (Figure 6c);
- fewer lightpath configurations are needed in the case of uniform cost because the consolidation has little impact on the processing cost (Figure 6d).

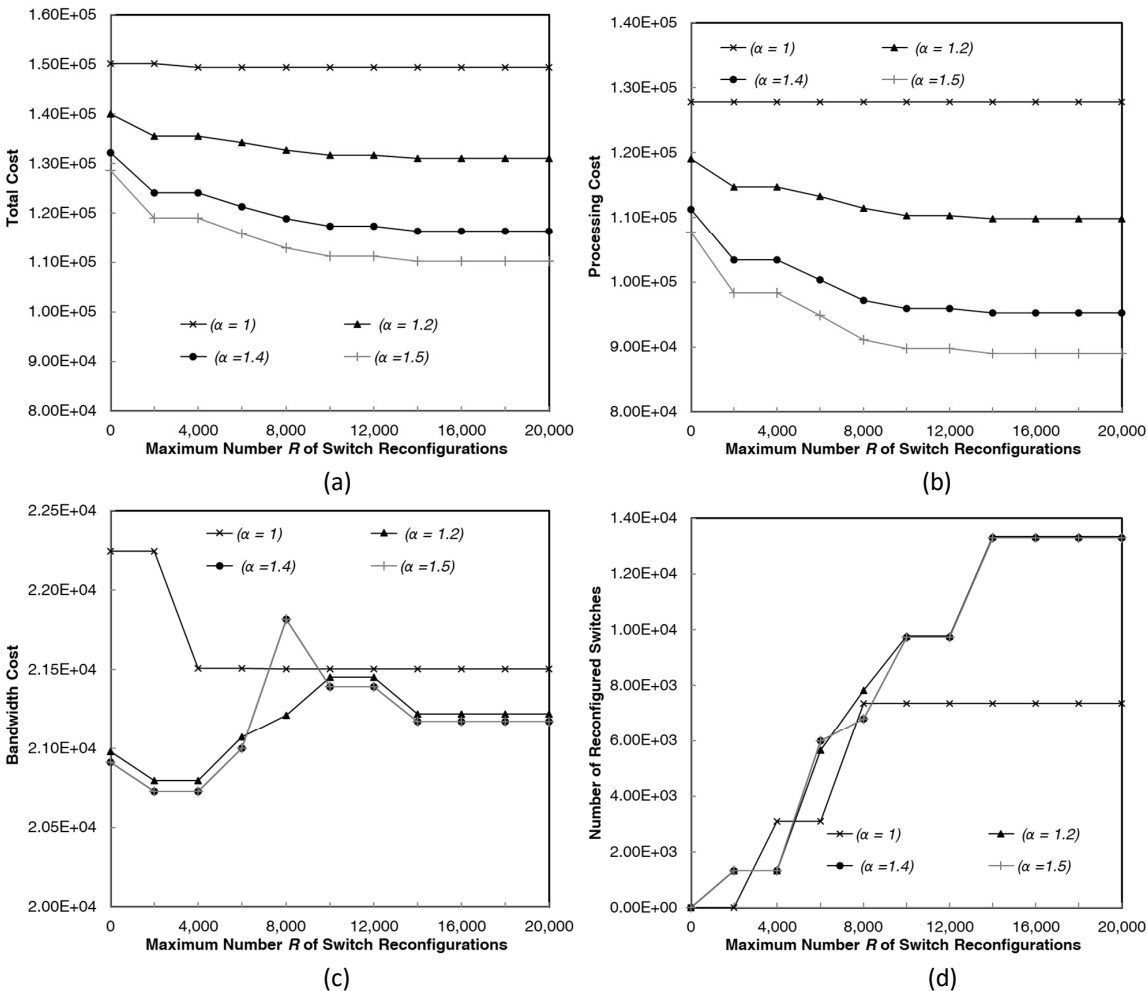

**Figure 6.** Total cost (**a**), processing cost (**b**), bandwidth cost (**c**) and number of reconfigured switches (**d**) as a function of the maximum number $R$ of switch reconfigurations. The offered traffic in the peak SI equals 800 Gbps. The cloud resource cost unbalancing parameter $\alpha$ is varied from 1 to 1.5. The Deutsche Telekom network topology is considered.

Finally, we provide some results for the USNET network topology of Figure 5 and when the unbalancing factor $\alpha$ is varied from 1 to 3. We illustrate the total cost and the percentage cost saving in Figure 7a,b respectively. Notice that the percentage cost saving is evaluated for each value of $\alpha$ with respect to the cost achieved when no reconfigurations are applied. We can observe from Figure 7a,b how the reconfiguration is more effective for larger values of $\alpha$ that lead to an advantages in consolidating towards lower cost NFVI-PoP. In particular in the case of $\alpha = 3$ the percentage cost saving is 20% when a number of switch reconfigurations larger than 10,000 is allowed.

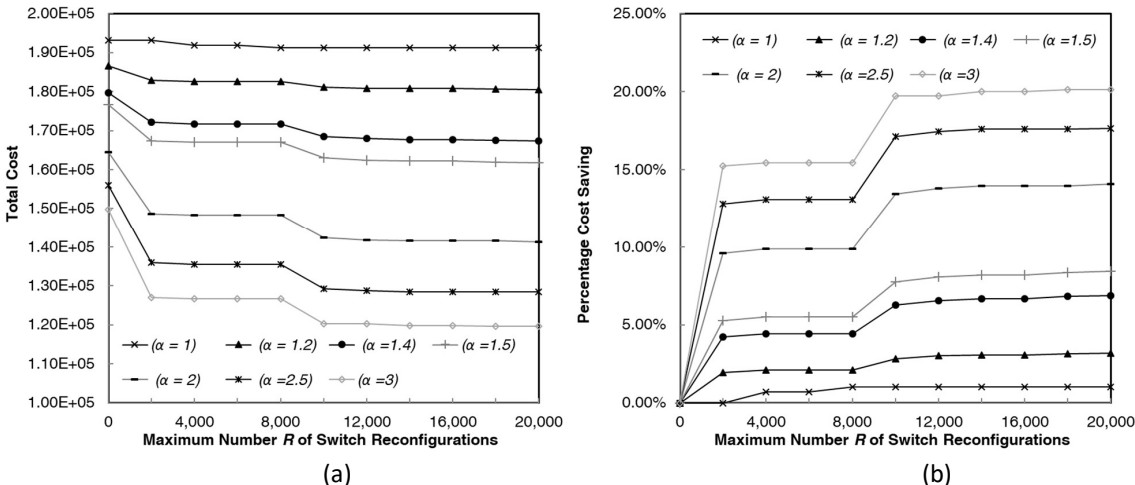

**Figure 7.** Total cost (**a**) and percentage cost saving (**b**) as a function of the maximum number $R$ of switch reconfigurations. The offered traffic in the peak SI equals 800 Gbps. The cloud resource cost unbalancing parameter $\alpha$ is varied from 1 to 3. The USNET network topology is considered.

## 6. Conclusions

The main contribution of this work is the proposal of a heuristic for the evaluation of the cost saving in reconfigurable NFV/Optical environment in which the number of switch reconfigurations is limited. It has been shown how the heuristic provides performance near to the one the ILP problem with an error percentage lower than 25%. We have also shown how the introduction of a constraint on the number of switch reconfigurations impacts on the cost saving, especially in the case of high core cost unbalancing. Conversely fewer reconfigurable lightpaths are needed in the uniform core cost case where the VM migration are not needed to lower the processing cost.

The analysis carried out in the paper assume an exact knowledge of the daily traffic profile. As future work we have planned to apply the proposed reconfiguration policies when a traffic prediction is performed. In such a scenario the impact of the prediction on the cloud and bandwidth resource overprovisioning and underprovisioning will be evaluated.

**Author Contributions:** V.E. has organized. F.G.L. and T.C. have implemented the code for the performance evaluation.

**Funding:** This research received no external funding.

**Conflicts of Interest:** The authors declare no conflict of interest.

## Appendix A. Optimization Problem

In this section, we describe the ILP problem formalized for the comparison with the heuristic proposed in Section 4. The problem is inherited from [16] and aims to minimize the overall costs while determining in each stationary interval in which NVFI-PoPs to execute the VNFIs and which lightpath channels to be used to connect them. in this new formulation, the contributions of the total costs are twofold: (i) the rented cloud resources; (ii) the bandwidth associated to the lightpath.

Another inherited concept is the one of the path channel that aims to satisfy contiguity and continuity constraints in the optical routing problem. As in [16], we evaluate off-line the available set $C_{\bar{p},q}$ of path channels that are associated to a given physical path $\bar{p}$ and given number $q$ of Frequency Slots (FSs).

The formalized problem is subject to two additional constraints that concern switch reconfiguration that occurs when a virtual link is mapped on a different lightpath with respect the previous SI.

The first one is the following:

$$\rho_{e\bar{e}}^{(h)} \geq \delta_{\bar{e}\bar{p}} \Big( \sum_{q\in[1..Q_{max}]} \sum_{c\in C_{\bar{p},q}} y_{ec}^{(h)} - \sum_{q\in[1..Q_{max}]} \sum_{c\in C_{\bar{p},q}} y_{ec}^{(h-1)\,\text{mod}\,N} \Big)$$

$$\forall e \in L; \forall \bar{e} \in \bar{L}; \ \ \forall h \in [0..T-1] \tag{A1}$$

wherein $\rho_{e\bar{e}}^{(h)}$ is a binary variable that represents the reconfiguration of a virtual link $e$ on the physical link $\bar{e}$ during the $h$-th SI, $\delta_{\bar{e}\bar{p}}$ is a binary parameter equal to 1 if the physical path $\bar{p}$ contains the physical link $\bar{e}$, and $y_{ec}^{(h)}$ is the binary variable that assumes value 1 if the virtual link $e$ is mapped on the path channel $c$ during the $h$-th SI. The constraint (A1) establishes that a virtual link $e \in L$ has been moved in the $h$-th SI when it has been mapped on $\bar{e} \in \bar{L}$ in the $h$-th SI but it does not in the $((h-1)\,\text{mod}\,N)$-th SI. Therefore this constraint establishes if a switch configuration occurs.

The second constraint introduced in the formalization is related to the the maximum number $R$ of switch reconfigurations to be applied to the elastic optical network during the cycle-stationary period and could be expressed as follow:

$$\sum_{e\in L} \sum_{\bar{e}\in\bar{L}} \sum_{h\in[0..T-1]} \rho_{e\bar{e}}^{(h)} \ \leq \ R \tag{A2}$$

As aforementioned, the optimization problem is based on the minimization of an objective function characterizing the total cost $C_{tot}$, that is:

$$min \ \ C_{tot} = C_{CR} + C_{BW} \tag{A3}$$

where $C_{CR}$ and $C_{BW}$ are the cloud and bandwidth resource cost respectively. If $T_s$ is the duration of a SI, the rented cloud resources cost $C_C R$ could be expressed as follow:

$$C_{CR} \ = \ T_s \sum_{h=0}^{T-1} \sum_{\bar{v}\in V_{NP}} c_{\bar{v}}^{core} \sum_{v\in V_{VI}} x_{v\bar{v}}^{(h)} N_v^{(h)} \tag{A4}$$

wherein $c_{\bar{v}}^{core}$ is the cost for renting a core in the NVFI-PoP $\bar{v}$ of the set $V_{NP}^{-}$, $x_{v\bar{v}}^{(h)}$ is a binary variable that assumes value 1 if the VNFI $v$ is mapped on the NFVI-PoP $\bar{v}$ during the $h$-th SI, and $N_v^{(h)}$ represents the number of cores that need to be allocated for the VNFI $v$ during the $h$-th SI.

The second component of the total cost $C_{BW}$ is the bandwidth cost associated to a lightpath and it is given by:

$$C_{BW} \ = \ T_s c_b \sum_{h=0}^{T-1} \sum_{e\in L} \sum_{\bar{p}\in\Psi} \sum_{c\in C_{\bar{p},e}^{(h)}} q B_{FS} y_{ec}^{(h)} L_{\bar{p}} \tag{A5}$$

wherein $c_b$ is the bandwidth cost unit, $B_{FS}$ is the bandwidth (Hz) of a Frequency Slot (FS), $q$ is the number of FSs of the path channel $c$ and $L_{\bar{p}}$ is the length of the network path $\bar{p}$.

Ultimately, it should note it is simple to demonstrate that the complexity of above ILP problem is NP-hard, thus to obtain significant results in a polynomial time an heuristic approach is needed.

## Appendix B. Formal Description of ONRCRA

More details on the Label Setting method [17] are given in this Appendix.

Given a directed graph which has costs associated to nodes and weights associated to arcs, the objective of the algorithm is to find the path with minimum cost from source node to destination node, with the constraint of not exceeding a maximum value of weight $W$.

The algorithm uses in each node a set of label, each one identifying a path from source node to that node and consisting of a pair of numbers representing the cost and the weight of the corresponding path.

If $I_i$ represents the index set of labels on node $i$, for each label $k \in I_i$ there is a corresponding path $P_i^k$ identified by the label $(W_i^k, C_i^k)$. For path $P_i^k = (j_0, j_1, \ldots, j_{n_i^k=1})$ weight and cost can be easily calculated by $W_i^k = w(P_i^k) = \sum_{m=1}^{n_i^k-1} w_{j_{m-1}, j_m}$ and $C_i^k = c(P_i^k) = \sum_{m=1}^{n_i^k-1} c_{j_{m-1}, j_m}$ where $w_{j_{m-1}, j_m}$ and $c_{j_{m-1}, j_m}$ are the weight and the cost of the edge $(j_{m-1}, j_m)$ respectively. Starting with a label only on the first node, at each iterations the algorithm extends all the labels extending the corresponding path along all outgoing arcs. If $\delta^+(i)$ is the set of outgoing arcs from node $i$, considering each arc $(i, j) \in \delta^+(i)$, a label $(W_i^k + w_{ij}, C_i^k + c_{ij})$ is saved on node $j$ only if $W_j^k = W_i^k + w_{ij} \leq W$, and for each existing label $l$ on node $j$ is verified that $W_j^k \leq W_i^l$ and $C_j^k \leq C_i^l$ (we will refer to this last property saying that a label added to a node is *efficient*).

When all the efficient labels have been generated, the algorithm stops and the solution path is the one idenfied by the label on destination node having minimal cost.

The pseudo-code of the algorithm is given in Algorithm A1 where $N$ is the set of nodes of the graph, $L_i$ denotes the set of labels in node $i$ and $T_i \subseteq I_i$ demote the index set of the labels on node $i$ which have been handled.

---

**Algorithm A1** THE LABEL SETTING ALGORITHM

---

1:   *Step 0*: *Inizialization*
2:   Set $L_s = \{(0, 0)\}$ and $L_i = \varnothing$ for each $i \in N \setminus \{s\}$.
3:   Initialize $I_i$ accordingly for each $i \in N$.
4:   Set $T_i = \varnothing$ for all $i \in V \setminus \{s\}$.
5:   *Step 1*: *Selection of the label to be handled*
6:   **if** $\cup_{i \in N}(I_i \setminus T_i = \varnothing)$ **then**
7:     STOP: all efficient labels have been generated
8:     **else** choose $i \in N$ and $k \in I_i \setminus T_i$ so that $W_i^k$ is minimal.
9:   **end if**
10:  *Step 2*: *Handling of label* $(W_i^k, C_i^k)$
11:  **for** all $(i, j) \in \delta^+(i)$ with $W_i^k + w_{ij} \leq W$ **do**
12:     **if** $(W_i^k + w_{ij}, C_i^k + c_{ij})$ is not dominated by $(W_j^l, C_j^l)$ for any $l \in I_j$ **then**
13:       Set $L_j = L_j \cup (W_i^k + w_{ij}, C_i^k + c_{ij})$.
14:       Update $I_j$ accordingly.
15:       Set $T_i := T_i \cup k$.
16:     **end if**
17:  **end for**
18:  **go to** Step 1.

---

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
