# Peer review of "Impact of the Maximum Number of Switching Reconfigurations on the Cost Saving in Network Function Virtualization Environments with Elastic Optical Interconnection†"

_applsci, doi:10.3390/app9235167_

Round 1

Reviewer 1 Report

The topic of the manuscript fits the scope of the Journal; nonetheless, it requires extra efforts to improve its quality and presentation for the prestigious journal Applied Sciences. A set of comments are expounded hereafter.

- The manuscript is, in general, well organized and well written. However, there are diverse faults regarding the format of the document, as commented below.

“And” must be inserted between the names of the last two authors.

The titles of sections should not be capitalized.

Once an acronym has been defined, it is not necessary to repeat such definition. This issue is found for NFV, VNF, VM, and others, which are defined in the introduction and also in the second section.

The citation of figures within the text should be using the whole word figure, for instance (line 94) “As shown in Figure 1…”.

In line 273, “Ghz” must be replaced by “GHz”.

The format of tables must be revised according to the template of the Journal.

The sections Authors contributions, Acknowledgments and Conflicts of interest are missing.

Concerning the references, the format must be revised to follow the template concerning some aspects like the abbreviated name of the journals, names of authors, etc.

Reference number 16 lacks the year of publication, 1983.

- About the content of the manuscript, it covers a very interesting topic. The comments after a careful revision are the following:

A keyword to include would be “switch reconfiguration”, if the authors agree with the suggestion.

The contextualization of the proposal is well organized and states in a proper manner the scope of the presented work.

Regarding the Label setting algorithm, such contextualization would be enhanced if some recent research work applying such algorithm were included. This would highlight the relevance of the presented work. For instance, the following paper:

Li, Q.; Tu, W.; Zhuo, L. Reliable Rescue Routing Optimization for Urban Emergency Logistics under Travel Time Uncertainty. ISPRS Int. J. Geo-Inf. 2018, 7, 77.

The software packages used for the research should be mentioned. This information is useful for the interested reader.

The results are properly expounded.

A desirable comment in the Conclusions section deals with future works that the authors are considering on the view of their developed work. Indeed, mentioning some limitations of the research would be adequate.

As a conclusion of the revision, if all the described suggestions are addressed, the manuscript will reach a better presentation and scientific level, according to the prestigious journal Applied Sciences.

Author Response

We would like to thank you the reviewer for its valuable comments. We considered them to improve the quality of the paper.

Please see our responses in the attachment file.

Reviewer 2 Report

In this paper, the authors studied the problem of minimizing operation costs of an NFV environment with data centers interconnected via an elastic optical network. Specifically, the authors considered the constraints of the number of switch reconfiguration for VM migration between data centers. The problem is formulated as an ILP optimization problem and a heuristic algorithm is proposed to determine embeddings of SFCs with a given number of switch reconfiguration as a constraint. Numerical results were also reported in the paper for evaluating the effectiveness of the proposed algorithm.

The problem investigated in this paper in general – resource allocation (SFC embedding) for minimizing NFV cost -- is interesting and important. On the other hand, the following issues of the current manuscript need to be addressed in revisions.

The problem investigated in this paper in general – resource allocation (SFC embedding) for minimizing NFV cost -- is interesting and important. On the other hand, the following issues of the current manuscript need to be addressed in revisions.

The problem formulation as ILP optimization is not clearly described in a complete mathematical format. Appendix A only gives two constraints (A1 and A2) but not the objective function.

A key feature of the method proposed in this paper is the maximum number of switch reconfiguration. However, it is not clear how such a maximum number can be determined so that a meaningful constraint for minimizing NFV cost can be obtained.

The idea of taking the maximum number of switch reconfiguration as a constraint needs to be further elaborated and justified. Why should we limit the number of switch reconfiguration if more reconfigurations may cause even lower cost in an NFV environment? It is true that there is a cost associated with switch reconfiguration, but it is also possible that more cost reduction may be enabled by such reconfigurations.

As a research article, the literature review part of the manuscript needs to be enhanced (probably with a dedicated section for related work) in order to thoroughly reflect the state of the art thus highlighting the advantages of the proposed approach. The research of this paper in general is related to federated management of network-cloud resources so the reviewer encourages the authors to refer the following recently published paper “Network Cloudification enabling Network-Cloud/Fog Service Unification: State of the Art and Challenges” by Q. Duan and S. Wang.

It would be better to have a table to list all the notations defined in subsection 3.1

It could be helpful to readers for understanding the algorithm if both procedures (NORR and ONRCRA) can be presented in pseudo-codes.

Author Response

We would like to thank you the reviewer for its valuable comments. We considered them to improve the quality of the paper.

Please see our responses in the attached file.
